# Brief mindfulness coaching enhances selective attention in medical scientists: A pilot study

Satish Jaiswal[1]*, Jason Nan[1], Suzanna R. Purpura[1], James K. Manchanda[1], Iris Garcia-pak[2], Dhakshin S. Ramanathan[1,2,3,4], Dawna Chuss[5], Deborah T. Rana[6,7], Ellen Beck[6,7], Paul A. Insel[2,8,9], Neil C. Chi[2,9], David M. Roth[5], Hemal H. Patel[5,10], Jyoti Mishra[1,4]*

1 Neural Engineering and Translation Labs, Department of Psychiatry, University of California, San Diego, La Jolla, California, United States of America, 2 Medical Scientist Training Program, University of California, San Diego, La Jolla, California, United States of America, 3 Department of Mental Health, VA San Diego Medical Center, San Diego, California, United States of America, 4 Center of Excellence for Stress and Mental Health, VA San Diego Medical Center, San Diego, California, United States of America, 5 Department of Anesthesiology, University of California, San Diego, La Jolla, California, United States of America, 6 Department of Family Medicine, University of California, San Diego, La Jolla, California, United States of America, 7 Herbert Wertheim School of Public Health and Human Longevity Science, University of California, San Diego, La Jolla, California, United States of America, 8 Department of Pharmacology, University of California, San Diego, La Jolla, California, United States of America, 9 Department of Medicine, University of California, San Diego, La Jolla, California, United States of America, 10 VA San Diego Medical Center, San Diego, California, United States of America

* s2jaiswal@ucsd.edu

## Abstract

Medical scientists have dual commitments to clinical care and research efforts. Such commitments can create hectic and stressful work schedules, which may impact on well-being and cognition. In this study, we tested the hypothesis that brief mindfulness coaching (three 1.5 hour online group sessions over 12 weeks) can benefit medical scientists. We conducted a waitlist-controlled intervention study (n = 43) with M.D./Ph.D. preclinical or graduate students and post M.D./Ph.D. trainees/faculty. Assessments of quantitative outcomes included self-reports of burnout, mindfulness, self-compassion, and well-being, as well as objective neuro-cognitive assessments. The results showed no effect of intervention on self-report measures. However, amongst cognitive measures, selective attention performance was significantly improved following the intervention (bias corrected effect size, Hedges' g = 1.13, p = 0.005). Extent of improvement in selective attention correlated with suppression of visual alpha oscillations – a neural marker for distractibility – measured using electroencephalography (EEG) (r = −0.32, p < 0.05). Qualitative feedback showed that after receiving the intervention, participants in both study arms equally rated the overall experience as "very good" (3.70 ± 0.98 out 5). They also appreciated that the intervention emphasized healthy lifestyle behaviors, and contributed to mindfulness, compassion, and a sense of community. A majority (57%) of the participants reported that they expect to change their well-being related behaviors because of the intervention.

**Data availability statement:** The dataset generated and analyzed in the current study is available from the datadryad.org repository: https://datadryad.org/dataset/doi:10.5061/dryad.zpc866tfb.

**Funding:** This work was supported by a grant from the National Institutes of Health, NIH/NIGMS 1T32GM121318-01 (DMR, HHP), involved UCSD-MSTP trainees supported by NIH/NIGMS 5T32GM007198, and a Research Career Scientist Award from the Veterans Administration (BX005229 to HHP). The funders had no role in study design, data collection and analysis, decision to publish, or preparation of the manuscript.

**Competing interests:** The authors declare no conflict of interest.

**Abbreviations:** ART, Attention Restoration Theory; DP, Depersonalization; EE, Emotional Exhaustion; EEG, Electroencephalography; ERS, Event-Related Synchronization; ERSP, Event Related Spectral Perturbations; ITI, Inter-Trial Interval; LSL, Lab Streaming Layer; MAAS, Mindful Attention Awareness Scale; MBI, Maslach Burnout Inventory; MSTP, Medical Scientist Training Program; NEAT Labs, Neural Engineering and Translational Labs; PA, Personal Accomplishment; ROI, Regions of Interest; SBL, Sparse Bayesian Learning; SWEMWBS, Short Warwick-Edinburgh mental Well-being Scale; UCSD, University of California San Diego; WM, Working memory.

Overall, this study suggests the utility of brief mindfulness coaching to improve selective attention skills in medical scientists and that more needs to be done to enhance subjective well-being in this healthcare workforce.

**Trial registration:** The study was registered in the International Standard Randomized Controlled Trial Number Registry (ISRCTN16736293) at https://www.isrctn.com/.

## 1. Introduction

The path of training to become a physician-scientist can be a long and potentially isolating journey. Such a career aims to bridge biomedical research and clinical care [1]. It is a unique role to learn from patient experiences that inspire new research questions and to use scientific discovery to develop new diagnostics and therapeutics. The training in the U.S. often consists of pursuit of a dual-degree program, M.D./Ph.D., sometimes in a NIH-supported Medical Scientist Training Program (MSTP). Such programs transpire over 8–9 years before graduates undertake clinical residencies [2]. Not only do these trainees toil through the rigorous workload of both medical school and graduate school, but their training integrates those environments and involves pressure to be efficient and accelerate through dual degree training. Transient states of belonging with peers can promote feelings of isolation among M.D./Ph.D. students, during post-M.D./Ph.D. training and for junior faculty [3]. For M.D./Ph.D. trainees, transitions between medical school and graduate school can be particularly stressful and are challenging for M.D.s who undertake laboratory efforts [4–7]. Physician-scientists in training may lack a supportive community and are at risk for poor well-being and burnout, a particular problem for individuals from groups underrepresented in the physician scientist community [8]. Physicians have a high rate of burnout, defined as "workplace stress that has not been successfully managed" [9]. Some common symptoms of burnout are: feelings of energy depletion or exhaustion; increased mental distance from one's job, or feelings of negativism or cynicism related to one's job; and reduced professional efficacy. This decline is evident through slower reaction times, increased error rates, or greater work fluctuations, ultimately leading to reduced work efficiency [10,11]. Based on prior reports on job demands and wellbeing, our theoretical assumption is that a high-pressure professional environment may affect the attentional capabilities and broader cognitive capacity of the physicians [11,12].

The restoration of this reduction in attentional capabilities can be explained by the attentional-resource or energetic-capacity model [11] that assumes depletion of a hypothetical reservoir of resources during work and recovery during rest. This model is based on Kaplan's attention-restoration theory (ART) [13], which emphasizes that "recovery is not only a function of leisure time but depends on the context where rest takes place." ART is a psycho-sociological theory that explores the dynamics of recovery and stress within real-world contexts of work and leisure. Hence, in the current study we propose an intervention program that would equip physicians with

skills, to use their period of rest actively to foster objectively measurable improvements in attentional or cognitive performance, as well as nurture subjective well-being.

Well-being can be defined as a positive state experienced by individuals, primarily focused on how people evaluate their lives, ranging from momentary moods to overall life satisfaction judgments [14]. Burnout rates in medical professionals are currently estimated at 44%, much higher than in other professions [15,16]. Moreover, the COVID19 pandemic exacerbated distress in healthcare workers and has been associated with dropout from training and reduction of work hours [17,18], which ultimately impacts the entire healthcare system. A recent study on medical trainees demonstrated the link between the constructs of burnout and well-being, reporting occurrence of burnout when the well-being reservoir was depleted [19]. Hence, a need exists for interventions that can help reduce burnout and promote well-being, especially for individuals training as medical scientists [20].

Research has shown that group-based wellness interventions, such as group counseling and psychotherapy, improve well-being and cognition, likely benefitting from shared experiences that facilitate cohesion and interaction among the group members [21–27]. In the past three decades, mindfulness-based interventions have received attention for alleviating stress and burnout symptoms [28–31]. However, it can be challenging for physicians to accommodate their schedules to participate in traditional mindfulness trainings, which usually last up to 8 weeks, requiring 2.5 hours per week for didactics and practice [32]. Also, having a fixed routine and coordinating in-person with a mindfulness coach can be a logistical challenge.

Given the need for implementing time-efficient well-being interventions for clinicians, particularly clinician scientists, we hypothesized that a digital intervention approach that integrates brief, online group mindfulness sessions led by expert coaches might be a useful way to enhance well-being. The intervention offered three coaching sessions, each for 1.5 hours, over one academic quarter (about 3 months), in addition to encouragement to practice at home. In each session the coaches conducted focused mindfulness exercises and led an interactive discussion on health, mindfulness, well-being, self-compassion and resilience, and leadership training. The group intervention was scheduled to accommodate the preferred times of the program participants. We evaluated change on self-report scales of burnout, mindfulness, self-compassion and well-being and compared outcomes to a control group that did not receive the intervention. Such self-report measures have been used in other mindfulness-integrated intervention studies [33–35], and to test mindfulness programs in physician cohorts [36–38]. For example, a study on physicians (n = 19) [36] demonstrated that video-module-based mindfulness intervention could significantly reduce the stress and symptoms of burnout, such as emotional exhaustion, but the study lacked a control group. Additionally, a randomized controlled trial (RCT) of mindfulness in physicians (n = 33) [37] demonstrated the improvement in burnout symptoms and enhanced mindfulness. Yet, these studies only focused on self-report measures. Subjective measures offer the advantage of providing insight into participants' personal assessments and feelings; however, they are susceptible to biases such as demand characteristics or tendencies toward certain self-presentation styles [39]. To mitigate these biases, objective measures, including cognitive function measures and their associated neural mechanisms, can offer a more comprehensive understanding of the intervention program's effectiveness. Moreover, given the evidence gap on objective outcomes in this literature, here we also sought to measure objective benefits of the intervention, specifically whether it enhances cognitive functioning abilities and engenders neuroplasticity. Thus, electroencephalography (EEG)-integrated objective cognitive assessments are a major strength and novelty of this study.

Cognitive abilities are fundamental to daily life functioning. There are several reports demonstrating that mindfulness-based interventions improve components of cognition such as attention and working memory [40–43]. Also the process of meditation itself encompasses a wide range of cognitive components during the practice itself such as selective attention, distractor suppression and conflict monitoring [44]. However, to the best of our knowledge there is no research on physician trainees that has assessed the modulation of cognitive functions driven by mindfulness intervention. Hence, here we implemented a scalable platform, the Brain Engagement (*BrainE©*) tool developed for free academic use by the Neural Engineering and Translation Labs (NEATLabs), for measuring core aspects of cognition. The *BrainE©* tool delivers standard cognitive assessments synchronized with neural recordings using EEG [45,46]. We have validated the reliability of

*BrainE©* and shown its utility in measuring cognitive changes across the lifespan [45,47–52]. We have also demonstrated its relevance in predicting mental health, well-being and burnout [53–61]. Here we deploy the *BrainE©* platform to measure post vs. pre-intervention changes in the mindfulness intervention group and compare outcomes with repeat assessments in a control group that controlled for practice effects of repeat assessments. Specifically, we investigated whether the intervention can alter foundational cognitive abilities of selective attention, working memory, and interference processing of both non-emotional and emotional distractors. It is important to study different facets of cognition in this context as it has been postulated that mindfulness training may variably impact different forms of cognition depending on several factors, such as style of practice, target cohorts, coach characteristics, and environmental factors [62,63].

Multidimensional cognitive assessments are also justified given that a recent metanalytic study reported that burnout may impair memory, attention and executive functioning and recommended mindfulness training as a protective remedy [64]. Our theoretical assumption is that the mindfulness training offered to physician trainees may facilitate improvements in present-moment-awareness and focus by cumulative practice over the intervention period that would be reflected in improved cognitive outcomes as well as concurrent neuroplasticity [65,66]. Furthermore, there is very limited literature exploring the impact of burnout on brain neuroplasticity among physicians [54,67,68]. EEG measures can provide insights into neural plasticity induced by the intervention [69–73], which has not been applied in the context of interventions for the healthcare community apart from a couple of recent studies [54,74]). In summary, we hypothesize that brief mindfulness training may equip physicians with mental skills that facilitate improvement of wellbeing, general cognition as well as neural changes that underlie those cognitive improvements.

## 2. Materials and methods

### 2.1. Participants

A total of 43 physician-scientists participated in the study (mean age: 28.95±4.36 years, range: 23–43 years, 42% female). Most (33 of 43 or 77%) participants were recruited from the University of California San Diego Medical Scientist Training Program (UCSD-MSTP); 10 of 43 (or 23%) had graduated from M.D./Ph.D. programs and were UCSD physicians/junior faculty at UCSD. Recruitment occurred via announcement at an annual retreat as well as by email. All participants provided written informed consent for study participation in accordance with the Declaration of Helsinki, and the UCSD Institutional Review Board approved all experimental procedures.

Inclusion criteria were current enrollment in the UCSD MSTP program or having graduated from an MD-PhD program. Data collection and intervention occurred from Fall 2021 (13th October 2021) through Summer 2022 (15th June 2022), where each quarter was about 3 months long. Participants provided demographic information at the beginning of the study. Of the 43 participants, 22 were randomly assigned to the mindfulness intervention group (18 current MSTP trainees and 4 M.D./Ph.D. graduates) and 21 were assigned to a control group (15 current MSTP trainees and 6 M.D./Ph.D. graduates). Fig 1, is a consort flowchart that provides an overview of how participants were enrolled, allocated and included in analysis.

The study was registered in the International Standard Randomized Controlled Trial Number Registry [75]. The mindfulness intervention group received the intervention during Fall-Winter quarter, while the control group received no intervention during this time. The control group eventually received the intervention during Spring-Summer quarter. All participants completed self-report and neuro-cognitive assessments before and after the first wave of the intervention, i.e., Fall-Winter quarter. At the end of Summer 2022, all participants in both groups completed a feedback survey. Experimenters and data analysts were blind to participants' assigned conditions.

No data were missing for self-report assessments. Two participants in the control group were missing cognitive and neural data. One participant in the mindfulness intervention group had corrupted neural data that could not be analyzed, hence, was marked as missing. Six participants (5 from the mindfulness intervention group and 1 from the control group) had missing data in the final qualitative survey.

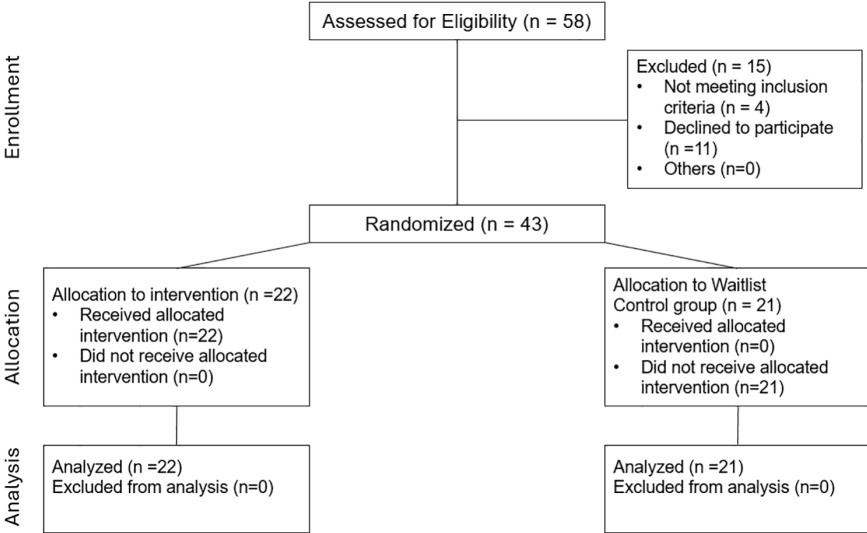

**Fig 1. Consort flowchart.** Overview of how participants were enrolled, allocated and included in analysis in the study.

## 2.2. Sample size and power

The sample size within each group (n = 22 intervention, n = 21 control) was powered to detect medium effect size pre vs. post difference (Cohen's d > 0.6) using t-test, with beta power of 0.8 and alpha level of 0.05. Between-group mean pre vs. post session differences met criteria for investigating only large effect size outcomes (Cohen's d > 0.8) by t-test at a beta power of 0.8 and alpha level of 0.05. Effect sizes were calculated a priori using the G*Power software (Faul et al. 2007), and reported as Hedges' g values that correct for the upward bias that can occur in Cohen's d when estimating effect sizes from small samples [76,77]. We acknowledge that our small sample size study can create challenges in interpretation of non-significant findings, as it would be unclear whether the absence of significance reflects true null effects or it is merely insufficient statistical power given we are not powered for d < 0.6 within-group effects or d < 0.8 between-group effects. Nonetheless, the findings from the current study may serve as a pilot template for a future larger sample replication study.

## 2.3. Intervention

The intervention was delivered via Zoom with group mindfulness peer mentoring sessions always led by two psychological health faculty as coach mentors, who did not know the participants prior to intervention. The rationale for the intervention was based on Kaplan's ART [13] that integrates the dynamics of recovery and stress within the real-world contexts of work and leisure, hence, the intervention program encompassed teaching skills that have real-world applications. In each session, the coaches conducted mindfulness exercises and led an interactive discussion on health, well-being, resilience, leadership training, and self-compassion as these topics relate to real life. For example, coaches taught that all activities or daily actions offer opportunities to be in practice by bringing awareness to them, e.g., 'washing dishes meditation' and informal practice in our daily life by bringing mindful awareness to whatever we may be doing in any given moment.

**Coaching Logistics:** For the coaching sessions, participants were divided into sub-groups by career stage: pre-clinical MSTP students, MSTP graduate students, or clinician/junior faculty, so that there was an equivalent number of participants of 4–8 per sub-group allowing for adequate interaction time between participants, and also so that participants could easily connect with each other's challenges given they were all in the same career-stage in a coaching session, making

for more cohesive sessions where everyone felt similarly engaged. The coaches led three sessions (each 1.5 hours) spread over one academic quarter for each sub-group. The group meetings were held virtually to maximize accessibility and comfort for the participants and were scheduled per the participants' mutual time preferences. Participants were asked to turn on their videos for the duration of the sessions and to share personal opinions and life experiences. In these sessions, the primary goal for the coaches was to listen attentively and promote well-being. They also aimed to validate and acknowledge challenges that participants faced as physician-scientists in training or as early career faculty. Different mindfulness tools and perspectives were suggested and discussed by the facilitators and by the participants themselves to address work-life challenges. The participants were encouraged to share their sources of support (e.g., exploring sources of strength, stress management techniques, and spending time with loved ones). Brief self-compassion exercise was offered and taught at each session such as soothing and supportive touch [78], and walking meditation modeled after the description provided in '*Wherever You Go There You Are*' [79]. The participants also discussed potential changes to the physician-scientist environment that would facilitate well-being, if implemented. At the end of each session participants were encouraged to have a tool or thought that was discussed and resonated with them, and to prepare to share one at the next session.

**Practices and Activities:** We opened and closed each session with a practice that we encouraged participants to integrate in their daily lives. These practices built from simpler, basic foundational practices to more complex forms that required engaging interaction, and all selected practices had much utility in real life – both work and leisure activities. These included (1) Practices for starting and facilitating meetings; (2) Mindfulness and Self-Compassion; (3) Sharing Sources of Inspiration; (4) Sources of Strength Reflection; (5) Goal-setting; (6) Skills in humanistic communication and emotional regulation; (7) Tools for building connection with self; (8) Tools for fear management; and (9) Moral Injury, locus of control and ways to respond. These practices are further described below -

(1) Practices for starting and facilitating meetings: At the beginning of each session, we practiced and encouraged starting with a moment of gathering, briefly checking in, and setting a goal or intention for the meeting. We then discussed approaches to running meetings that create safe spaces for participants so that voices can be safely heard without danger of risk of retaliation. We discussed setting ground rules for this communication and healthy boundaries for maintenance of confidentiality. We encouraged developing mechanisms for people to share their ideas and concerns anonymously so that there is no fear of retaliation or concern for power inequities. These practices are important so that safe spaces can be built and maintained, and trust can be fostered [80].

(2) Mindfulness and Self- Compassion: Walking meditation is mindful meditation focusing on the present moment and the sensory experience of walking. Similarly, all daily life actions offer opportunities to be in practice by bringing awareness to them, e.g., 'washing dishes meditation.' [80]. Both walking and standing meditations were practiced with participants [79], and with the invitation to express peace in every step [81] and to walk in beauty wherever you are (Dine/Navajo saying) [82]. This involves paying attention on purpose to the cycle of walking, of shifting, lifting and placing, as well as the sensation of walking on the soles of the feet, in particular, and walking with or without a destination (a walk to a patient, a walk in the lab) and then standing and paying attention to breathing, and the shifting of balance rooted as a tree with roots beneath one's feet spreading beneath the "ground" below. We discussed these as grounding exercises and ways to remain anchored yet be flexible throughout the training as a physician-scientist and beyond into the years as a faculty member.

More informal practices were also shared based on the challenges and issues that the participants brought up. Self-compassion and self-kindness (extending compassion to oneself in moments of perceived inadequacy, self-criticism, external or internal feedback, failure, or general suffering) were discussed. To prepare to apply compassion and kindness to ourselves, we first identified a deep wise part within ourselves, the part which offers perspective. We practiced strengthening that part and sending ourselves kindness and compassion after reflecting on and bringing to mind a moment of

challenge, and called forth how we may offer this kindness and support to a friend and directed this same support and kindness to ourselves. The practice *Soften, Soothe, Allow* was then practiced as a form of mindful awareness of emotion expressed in the body [83]. By applying body awareness in moments where we experience a difficult emotion, we may note a release in intensity, and connect body and mind – allowing the emotion to be noted and held by cultivating non-judgmental awareness and soothing the places of discomfort with gentle or soothing touch.

(3) Sharing Sources of Inspiration: We invited participants to bring and share inspirational quotes, teachings, and music which have nourished them over the years. The goal of this activity is community and trust-building in the group, as well as mutual inspiration. We used several quotations to help prompt this process including the following:

| |
|---|
| "We must be the change we wish to see in the world." – Mahatma Gandhi |
| "We must emancipate ourselves from mental slavery, only we can free our minds."- Marcus Garvey, quoted by Bob Marley in the song, Redemption Song. |
| "We must not move the way fear moves us. We wake up each morning empty and frightened. Do not immediately go into the study. Take down a musical instrument." – Rumi |
| "Injustice anywhere is a threat to justice everywhere." – Martin Luther King, Jr. |
| "Si, se puede." – Dolores Huerta |
| "If I am not for myself, who will be for me. If I am only for myself, what I am I. If not now, when?" – Hillel |

The above quotes were chosen to represent diverse individuals across different cultures and time periods that would resonate with the diverse participants, and all quotes were selected to share powerful common themes of personal empowerment, courage, inner transformation as a catalyst for outer change, and collective and ethical responsibility within our communities. Participants brought favorite quotes, teachings, etc. and shared how the words came into their lives, their impact on their lives, and why these words are important to them.

(4) Sources of strength reflection: We invited participants to identify their sources of strength – practices that helped nourish them over the years. We taught ways to identify sources of strength – the things and practices that helped them feel connected and supported [84,85], and remain aware of these sources of strength and nourishment in times of challenge. Sources of strength may include music, reading, time with family, faith, dance, sports, etc. The practice of writing a 'Sources of Strength Prescription' can be practiced for self or others. It begins with identifying those activities and practices that help us stay connected with our deepest self and to practice them. We discussed that often in times of stress and illness, we let go of these practices just when we need them the most. Most of us go in our 'heads' into survival mode. Although we have learned how to survive and these survival behaviors have helped us in the past, often when stress or illness present themselves, over time, this survival attitude can disserve us. Learning to stay connected with and practice our sources of strength, in an ongoing way, and especially in times of stress was explored with participants. Each participant was asked to share how they might practice their sources of strength in their current life in a realistic fashion.

For a number of participants, time with and support from family and friends as well as time in nature were important to them. We discussed the concepts of 'family of origin' and 'family of choice'. In relation to nature, we talked about not only looking at and walking among the trees or by the sea, but actually practicing, for a moment, 'being the tree' or 'being the flower.' A traditional practice was also shared, of once a day, going outside, into nature, and 'pouring out our heart' and the rest of the day, practicing living life fully and 'being in joy.' We also discussed 'awe pauses', stopping for a moment, in the middle of a busy day, to pause and look at a tree, a photo of a child or grandchild, or listening to or singing a favorite song, to connect with a sense of wonder.

(5) Goal-setting: We addressed several tools for goal-setting, including a goal-setting activity in which one responds both what they seek for themselves and for the world [86]. We also invited participant to use a simple set of drawings to identify goals, obstacles and how to overcome them [87].

(6) Skills in humanistic communication and emotional regulation: Effective humanistic communication consists of three core skills: respect to all (unconditional positive regard), empathy, and self-awareness [88]. We encouraged the practice of self-aware communication and deep listening in the learning and sharing space of the sessions; here, we encouraged each person to recognize what they may be feeling and integrate this into their interpersonal interactions. We also talked about practices for effective communication within relationships, both in the personal and work setting and the importance of building trust.

(7) Tools for building connection with self: We discussed a paradigm of internal communication with mind, body, and spirit—where we interact with each aspect of ourselves, and, in which, spirit may be redefined as connection with one's core values and beliefs. The goal of this framework is to stay connected with one's core values, 'one's deepest wisdom' and perspective, and from this place to offer support and caring to our own mind and body.

(8) Tools for fear management: These tools included becoming aware that the fears we create often have positive goals to protect us, help us succeed, keep us safe, but have terrible technique, i.e., panic attacks, insomnia, anxiety, paralysis. We encouraged helping ourselves and others by having a 'conversation' with our fears, to help transform our fears into useful small inner messages and to help them become more manageable and less overwhelming.

(9) Moral Injury, locus of control and ways to respond: In the current healthcare practice and medical training environment, we discussed that often anxiety, exhaustion, and sadness may be appropriate responses to a system that is often destructive of the human spirit, and that sadness and concerns may actually be *healthy responses to a sick system*. We discussed the concept of locus of control and that ultimately, we had control over one thing – our attitude towards the event [89]. The concept of fierce compassion was introduced as a way to take a stand for what is called for to navigate change [90]. At the same time, we need to offer ourselves compassion while raising our hand to stop an unjust action encountered in the system, when these actions are not in alignment with the values that may have brought us to pursue a career as a physician-scientist. We also addressed the possible risks of such behaviors. Additionally, we discussed *Social Tai-Chi*, a practice of centering oneself and then choosing the next move, forward, back to the side, as we work toward a shared attention or goal, and discussed moving from being a passive ally to a becoming an active co-conspirator in the fight for justice [91].

Overall, we encouraged participants to identify practices that were nourishing to them and to modify and embrace them in a way that would allow them to integrate them into their daily lives. We encouraged participants to continue these practices in their daily lives in an ongoing and realistic way and to develop gentle reminders for themselves to continue to use them, e.g., a sticker on their computer, or a message on their phone. After the first session, participants shared some ongoing examples of bringing some of these practices into their workplace, during time with their children, and when opening the door to see a patient. Mapping of the intervention practices to study outcomes is summarized in Table 1 below.

Of note, at the beginning of the study, all 43 participants in the intervention and the waitlist group were each provided a Garmin Vivosmart® 4 watch (a wearable device) to passively monitor lifestyle factors (physical activity, heart rate, sleep quality and stress) via a smartphone app coupled to the Garmin watch. The Garmin watches were provided to all participants at the same time, so this digital intervention component did not differ between the intervention/waitlist groups. All notifications were turned off on these devices with no feedback provided to individuals based on the watch data. The participants could engage with the app and look at their own data, so the watch served to provide passive cues to support mindfulness engagement in both intervention and waitlist participants. But due to technical issues with the app the study team had no access to watch data regarding adherence or lifestyle fluctuations. Hence, only the timeline of the mindfulness coaching intervention period differed between the intervention/waitlist groups.

**Table 1. Putative mapping of intervention practices to study outcomes.**

| Intervention Practice | Study Outcome |
|---|---|
| 1. Practices for initiation | Mindfulness, Well-being; **Cognition**: Attention |
| 2. Mindfulness & Self-Compassion | Mindfulness, Self-compassion, Well-being; **Cognition**: Attention, Interference Processing, Emotion Regulation |
| 3. Sharing Sources of Inspiration | Well-being; **Cognition**: Attention, Working Memory |
| 4. Sources of Strength Reflection | Well-being, Burnout; **Cognition**: Interference Processing, Emotion Regulation |
| 5. Goal-setting | **Cognition**: Attention, Working Memory |
| 6. Skills in humanistic communication and emotional regulation | Mindfulness; **Cognition**: Attention, Emotion Regulation |
| 7. Tools for building connection with self | Mindfulness, Well-being, Burnout; **Cognition**: Attention, Working Memory |
| 8. Tools for fear management | Mindfulness, Self-compassion, Well-being, Burnout; **Cognition**: Attention, Emotion Regulation |
| 9. Moral Injury and locus of control | Self-compassion, Well-being, Burnout; **Cognition**: Attention, Working Memory, Emotion Regulation |

## 2.4. Assessments

These consisted of (1) self-report scales, (2) cognitive and neural assessments and (3) a terminal feedback experience survey. Self-report and neuro-cognitive assessments were performed at pre- and post-intervention for the mindfulness intervention group while the control group performed these repeat assessments 3 months apart without any intervention and hence controlled for practice effects. The terminal feedback survey was administered after both groups had received the intervention.

(1) Self-report assessments were completed electronically and included validated self-report scales of mindfulness: 14-item mindful attention awareness scale (MAAS) in the range of 1–6, where 1 = 'almost always', and 6 = 'almost never' [92]; well-being: 7-item Short Warwick-Edinburgh mental well-being Likert scale of 1–5, where 1 = 'none of the time' and 5 = 'all of the time'; (SWEMWBS) [93], self-compassion: 12-item self-compassion scale in the range of 1–5, where 1 = 'almost never', and 5 = 'almost always' [94]; and burnout: 22-item Maslach Burnout Inventory (MBI) on a six-point Likert scale (ranging from 0 = 'never' to 6 = 'everyday') [95,96].

(2) Neuro-cognitive assessments were deployed on the *BrainE©* platform coded in Unity with simultaneous EEG [45]. Participants visited the UCSD Neural Engineering and Translational Labs (NEAT Labs) for these assessments, which were delivered on a Windows-10 laptop. The Lab Streaming Layer (LSL) protocol was used to timestamp all stimuli and response events in all cognitive tasks [97]. Each session lasted ~40 minutes and consisted of four different cognitive tasks; each session additionally allowed for breaks between tasks to minimize fatigue. Tasks were run in the same order for all participants.

Fig 2 shows the stimulus sequence in each task. All four cognitive tasks had a standard trial structure of 500 ms central fixation "+" cue followed by a task-specific stimulus presented for task-specific duration and with a task-specific response window. This response window in each task was adaptive with a 3up-1down staircase scheme that maintains accuracy at ~80% and engages the user by avoiding ceiling performance [98,99]. An adaptive scheme also reduces practice effects that affect repeat assessment sessions. Further details of the adaptive scheme in each task are provided below.

Stimuli in each cognitive task were presented in a shuffled order across trials. Response in every task trial was followed by standard response feedback for accuracy as a smiley or sad face emoticon, presented 200 ms post-response

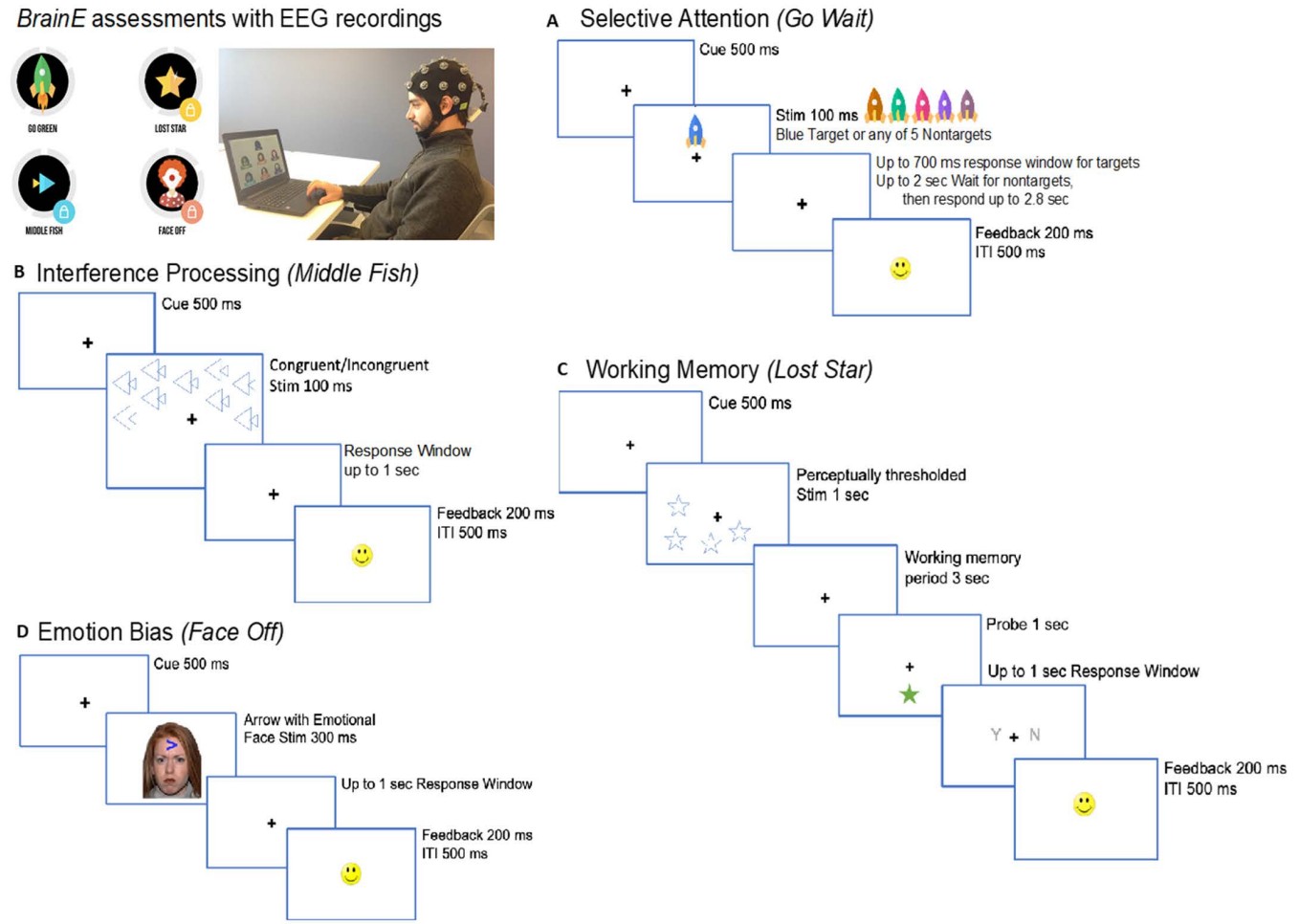

**Fig 2. Cognitive assessments delivered on the *BrainE* platform.** At top left, the *BrainE* assessment dashboard is shown with the wireless EEG recording setup. **(A)** Selective attention was measured on the 'Go' trials of the *Go Wait* task that required rapid and accurate responses to blue rocket targets. All other color rockets were non-targets on which responses were inhibited for up to 2 sec. **(B)** In the Flanker interference processing *Middle Fish* task, flanking fish may either face the same direction as the middle fish on congruent trials as shown, or the opposite direction on incongruent trials; participants were instructed to respond to the direction of the middle fish. **(C)** The visuo-spatial working memory task, *Lost Star*, was presented with perceptually thresholded stimuli and participants responded after a working memory period whether a probe star was positioned in one of the same locations as the prior test star stimuli. **(D)** The emotion bias task, *Face Off* presented neutral, happy, sad, or angry faces superimposed on an arrow, whose direction was discriminated by participants.

for 200 ms duration, followed by a 500 ms inter-trial interval (ITI). ITI jitter within tasks was not applied to keep the task administration rapid. At the end of each task block, participants received a percent block accuracy score with a series of happy face emoticons (up to 10) to promote engagement.

(2A) Selective Attention: Participants accessed a game-like task, *Go Wait* modeled after the standard test of variables of attention [100]. On each task trial, colored rockets were presented in the upper or lower central visual field. Participants were instructed to respond as rapidly as possible to blue colored rocket targets and wait to respond for 2 sec to distracting rockets of five other iso-luminant colors (shades of brown, teal, pink, purple). Iso-luminant colors were ensured using luminosity measurements in Photoshop. Luminance values for iso-luminant stimuli were 128, estimated as 0.30*R + 0.59*G + 0.11*B (where RGB are the Red/Green/Blue values of the chosen stimulus color). Post-fixation cue, a

target/non-target stimulus appeared for 100 ms duration. For the blue rocket targets, the initial response window was set at 700 ms that adapted on each trial in a 3up-1down scheme, i.e., the response window reduced −33 ms after correct trials and increased +100 ms after incorrect trials. One happy face emoticon followed correct trials. If response was very rapid within 100–400 ms, then two happy face emoticons were presented for feedback to reinforce fast and accurate responding [101]. For nontarget rockets, response times were not adaptive; participants waited for 2 sec at which time the fixation cue flashed briefly for 100 ms and then participants responded. Across two blocks, target and non-target trials were shuffled with 50% probability for 180 trials. Since task adaptivity was focused on target trials, processing speed on these trials was taken as the main selective attention outcome measure.

**(2B)** Interference Processing: Participants accessed the game-like task, *Middle Fish*, an adaptation of the Flanker assessment [102–104]. Post-fixation on each trial, participants viewed an array of fish presented in the upper or lower central visual field for 100 ms. On each trial, participants had up to a 1 sec response window to detect the direction of the middle fish in the set (left or right) while ignoring the flanking distractor fish that were either congruent or incongruent to the middle fish, i.e., faced the same or opposite direction to the middle fish. Response windows were adapted on congruent trials in a 3up-1down scheme (-33 ms after correct trials and +100 ms after incorrect trials) and incongruent trial response windows matched that of the previous congruent trial. The fish flanking the middle fish interfere with the discrimination of the direction of the middle fish (left/right), hence, the task assesses interference processing. Task trials were shuffled with congruent/incongruent distractors in 1:1 ratio in 96 trials over two blocks. Processing speed on all trials was monitored as the main outcome.

**(2C)** Working Memory: Participants accessed a game-like task, *Lost Star*, which was based on the visuo-spatial Sternberg task [105]. Post-fixation cue on each trial, participants viewed a spatially distributed test array of objects (i.e., a set of blue stars) for 1 sec. Participants were required to maintain the locations of these stars for a 3 sec delay period utilizing their working memory. A probe object (a single green star of 1 sec duration) was then presented in either the same spot as one of the original test stars, or in a different spot than any of the original test stars. The participant was instructed to respond whether the probe star had the same or different location as one of the test stars. 50% of task trials had the same probe star location as one of the test stars while 50% had different location, and presented in shuffled order. For each participant, we implemented this task at the threshold perceptual span, which was defined by the number of test star objects that the individual could correctly encode without any working memory delay. For this, a brief perceptual thresholding period preceded the main working memory task, allowing for equivalent perceptual load to be investigated across participants [103]. During thresholding, the set size of test stars increased progressively from 1−8 stars based on accurate performance where 100% accuracy led to an increment in set size; <100% performance led to one 4-trial repeat of the same set size and any further inaccurate performance aborted the thresholding phase. The final set size at which 100% accuracy was obtained was designated as the individual's perceptual threshold. Post-thresholding, the working memory task presented 48 trials over two blocks. Unlike the two previous tasks, this task was not speeded but instead adapted the working memory period in a 3up-1down scheme; if correct, the working memory period increased by +0.9 sec or if incorrect, it decreased by −0.3 sec. Both task item span and the length of the adapted working memory period were monitored as outcome measures.

**(2D)** Emotion Bias: Participants accessed the game-like assessment, *Face Off*, adapted from studies of attentional bias in emotional contexts [106,107]. [100] The task integrated a standardized set of culturally diverse faces from the NimStim database [108]. We used an equivalent number of male and female faces, each face with four sets of emotions: neutral, positive (happy), negative (sad) or threatening (angry), presented on equivalent number of trials in each task block. Post-fixation cue on each trial, participants viewed an emotional face with a superimposed arrow of 300 ms duration. The arrow occurred in either the upper or lower central visual field on equal number of trials. Participants responded to the direction of the arrow (left/right) within an ensuing 1 sec response window. For neutral emotion trials, this response window adapted in a 3up-1down scheme (-33 ms after correct trials and +100 ms after incorrect trials). All other emotion trials

followed the same response window as their previous neutral emotion trial. This task evaluates emotion bias (or interference) as the emotional faces interfere with the discrimination of the direction (left/right) of the arrow on which they are superimposed. Participants completed 144 trials presented over three equipartitioned blocks. Processing speed across trials was monitored as the main outcome.

Mapping of the intervention practices as they may influence the study outcomes is summarized in Table 1 below.

EEG data were collected in conjunction with all cognitive tasks using a 24-channel saline-soaked electrode cap with electrode locations as per the 10–20 system attached to a wireless SMARTING™ amplifier. Signals were acquired at 500 Hz sampling frequency at 24-bit resolution. The Lab Streaming Layer (LSL) protocol was used to time-stamp EEG markers and integrate cognitive markers [97]. Files were stored in xdf format.

(3) The end-of-year feedback survey was provided after all participants groups had undergone the intervention. The survey asked about participants' overall experience on a 1 (poor) to 5 (excellent) Likert scale. It also asked participants to provide a Yes/No response as to whether they expected future behavior change due to the intervention. Participants also were asked what they particularly liked/learned from the program and/or what could be improved.

## 2.5. Outcome measures and data analysis

All study measures were tested for normality using the Levene's test. The normally distributed data were evaluated with parametric repeated measure analysis of variance (rm-ANOVA) for group x session interactions, while non-normal data were evaluated using the non-parametric equivalent of ANOVA, i.e., Kruskal-Wallis (KW) test [109]. For any significant ANOVA results, post-hoc analyses were conducted within each group using either paired t-tests for normal data or the non-parametric equivalent Wilcoxon signed-rank test for non-normal data, and between-group effects in the ANOVA were further interrogated using either the two-sample t-test for normal data or otherwise the non-parametric Wilcoxon rank sum test was performed.

**2.5.1. Demographics.** Demographic characteristics of age, gender and ethnicity were compared between the two groups to confirm there were no group differences in these characteristics using the non-parametric Wilcoxon rank sum test for age comparisons and the $\chi^2$ (Chi-Square) test for gender and ethnicity comparisons.

**2.5.2. Self-report analyses.** For self-report assessments, we calculated scores on the mindfulness, well-being, self-compassion assessments by averaging the total score with respect to total corresponding number of items in these surveys. For the 3 burnout component scales, scores were calculated by summing responses on each subscale. Normality of distributions was checked using the Levene's test; we observed that all 3 facets of burnout, self-compassion, and some cognitive measures, i.e., working memory span and working memory period were not normally distributed, while all other measures had normally distributed scores. The rm-ANOVA and KW tests revealed that except selective attention processing speed none of the self-report measures nor any other cognitive measures showed group x session interactions. In the following Results section, we provide the pre and post session values of all measures in Tables 3 and 4, for self-report and cognitive measures, respectively. The detailed rm-ANOVA results are provided only for the selective attentions processing speed measure.

**2.5.3. Cognitive assessment analyses.** The data for all cognitive tasks were analyzed for the task-relevant outcome measure. For the selective attention, interference processing and emotion bias tasks, processing speed was calculated at both pre and post sessions as log(1/RT), where RT is response time in seconds; thus, longer RTs have less speed while shorter RTs have higher processing speed. For the working memory task, item span and working memory delay period length were both outcomes. Outliers >3 median absolute deviation (mad) from median were removed and all metrics were verified for normal distributions prior to statistical analyses. Rm-ANOVAs were used to interrogate group x session interactions. Additionally, false discovery rate (fdr) corrections were applied to account for multiple comparisons. Post-hoc analyses within each group used paired t-tests.

For the end of year feedback survey, overall experience ratings were compared between groups using Wilcoxon rank sum tests. Binary expected behavior change was analyzed using the Chi-Square test. Themes were extracted from qualitative responses and their proportional occurrence in each group were also compared using Chi-Square tests.

Effect sizes are reported for significant results as Cohen's d, 0.2: small, 0.5: medium, 0.8: large [110].

**2.5.4. Neural assessment analysis.** Neural data were analyzed for cognitive task(s) that showed any significant intervention-related cognitive performance effects. A uniform processing pipeline was applied to EEG data based on the LSL event markers. Data were pre-processed for computing event related spectral perturbations (ERSP) and for source localization. Data pre-processing utilized the EEGLAB toolbox in MATLAB [111]. EEG data were first resampled at 250 Hz and filtered in the 1−45 Hz range to exclude ultraslow DC drifts at <1Hz and high-frequency noise produced by muscle movements and external electrical sources at >45Hz. EEG data were average electrode referenced and epoched to stimulus onset based on the LSL timestamps, within the −1.0 to +1.0 sec event time window. The epoched data were then cleaned using the *autorej* function of EEGLAB, which automatically removes noisy trials (>5sd outliers rejected over max 8 iterations). EEG data were further cleaned by excluding signals estimated to be originating from non-brain sources, such as electro-oculographic, electromyographic or unknown sources, using the Sparse Bayesian learning (SBL) algorithm [112,113]. This algorithm localizes EEG data similar to low-resolution electromagnetic tomography (LORETA) [114] with sparsity constraints applied to protect against false positives that are not biologically plausible. SBL yields source configurations from a few active regions, akin to independent component analysis, and are maximally independent from one another. Data from non-brain sources are removed to obtain clean subject-wise trial-averaged EEG. We verified that for cleaned data, channel peak activity in individual participant data did not exceed 3 mad from average channel activity across all subjects (total n = 40).

For ERSP calculations, we performed time-frequency decomposition of the epoched data using the continuous wavelet transform (cwt) function in MATLAB's signal processing toolbox. Event-related synchronization (ERS) and event-related desynchronization (ERD) modulations were computed as baseline-corrected activity using the −750 ms to −550 ms time window prior to stimulus presentation as the baseline [115]. Outliers >3 mad activity level were removed from the final ERSP matrices. The cleaned data were then band filtered in the physiologically relevant theta (4–7 Hz), alpha (8–12 Hz), and beta (13–30 Hz) frequency bands and band-specific epoched activity was source localized using the SBL algorithm.

SBL's data-driven sparsity constraints reduce the effective number of sources considered at any given time as a solution, thereby reducing the uncertainty of the inverse solution. Thus, not only can higher channel density data yield source solutions, the ill-posed inverse problem can also be solved by imposing more aggressive constraints on the solution to converge on the source model at lower channel densities, as supported by prior research [116,117]. We also benchmarked the SBL algorithm to show that it produces evidence-optimized inverse source models at 0.95AUC relative to the ground truth [112,113], and that the regions of interest (ROI) estimates resulting from this cortical source mapping have high test-retest reliability (Cronbach's alpha = 0.77, p < 0.0001 [45]).

For the source space activations, ROIs were based on the standard 68 brain region Desikan-Killiany atlas [118] using the Colin-27 head model [119]. Artifacts remaining in source space were removed across all sessions and subjects using the > 3 mad criterion. We compared the change in source ROI activity between pre- and post-sessions using rm-ANOVA with Greenhouse–Geisser significance correction applied. We also confirmed that the source activity values were normally distributed. Spearman's correlations were used to analyze neural change and cognitive performance change associations across sessions.

## 3. Results

### 3.1. Group demographics

Table 2 shows demographics of both groups with regard to age, gender and ethnicity. There were no group differences in these demographics.

### 3.2. Self-report and cognitive results

Table 3 shows scores for the self-report scales in both groups and at both pre and post sessions. There were no baseline group differences. Notably, amongst burnout measures, only emotional exhaustion was moderately high (greater than mid-scale score of 18). MBI depersonalization was low and personal accomplishment was high and these two

Table 2. Summary of participant demographics. Mean±standard deviation data are shown for age in years, and gender and ethnicity are shown as percentage of total sample. Age was compared between groups using the non-parametric Wilcoxon Rank Sum test, and gender and ethnicity variables were compared between groups using Chi-square tests.

| Demographics | Intervention (n=22) | Control (n=21) | Group diff p-val |
|---|---|---|---|
| Age (years) | 28.27±3.68 | 29.67±4.96 | 0.407 |
| Gender n (%) | | | 0.700 |
| Male | 10 (45.45) | 12 (57.14) | |
| Female | 10 (45.45) | 8 (38.10) | |
| Ethnicity n (%) | | | 0.44 |
| Caucasian | 11 (50) | 9 (42.86) | |
| Black/African American | 1 (4.55) | 1 (4.76) | |
| Native Hawaiian or Other Pacific | 0 | 0 | |
| Asian | 3 (13.64) | 8 (38.10) | |
| Native American | 0 | 0 | |
| More than one ethnicity | 5 (22.73) | 2 (9.52) | |
| Other | 1 (4.55) | 1 (4.76) | |

Table 3. Summary of self-report outcome assessments from pre and post sessions (Intervention Group, n=22; Control Group, n=21). Mean±standard deviation data are shown for all variables. Effect sizes of between group differences (Hedge's g) with lower and upper 95% confidence intervals are also shown. MBI: Maslach Burnout Inventory included EE: Emotional Exhaustion, PA: Personal Accomplishment scores and DP: Depersonalization.

| Outcomes | Mindfulness Intervention | | Control | | Intervention Effect Sizes |
|---|---|---|---|---|---|
| | Pre | Post | Pre | Post | |
| Mindfulness | 3.47±0.66 | 3.34±0.85 | 3.13±0.75 | 3.31±0.74 | −0.50 [−0.63, 0.07] |
| Well-being | 3.25±0.68 | 3.37±0.62 | 3.32±0.56 | 3.32±0.66 | 0.20 [−0.27, 0.54] |
| Self-Compassion | 2.86±0.74 | 2.95±0.70 | 3.08±0.71 | 3.15±0.66 | 0.12 [−0.25, 0.38] |
| MBI EE | 26.95±13.46 | 25.68±12.21 | 24.05±9.97 | 23.05±12.15 | −0.03 [−6.48, 5.51] |
| MBI PA | 31.52±10.51 | 32.18±10.08 | 32.86±6.57 | 34.67±7.58 | −0.18 [−4.61, 2.52] |
| MBI DP | 5.67±4.17 | 6.09±4.16 | 9.29±7.74 | 7.57±7.19 | 0.46 [−0.80, 5.47] |

subcomponents did not indicate burnout. Between-group Kruskal-Wallis test comparisons on the post minus pre difference showed no effects for any of the measures of mindfulness, well-being, self-compassion and burnout (all p>0.05).

Table 4 shows cognitive performance outcomes in both groups at both pre and post sessions. Performance measures were compared using rm-ANOVAs; **significant results were fdr-corrected for multiple comparisons.** Processing speed on the selective attention task was the only measure with a significant group x session interaction ($F_{2,39}$=12.71, p=0.005 fdr-corrected, 95% CI [0.01 to 0.05], bias corrected effect size (Hedges' g) = 1.13); there were no main effects (p>0. 05). Within-group post-hoc testing showed that selective attention processing speed was improved at post- relative to pre-session for the intervention group (p=0.019) and worsened for the control group (p=0.022) (Fig 3).

### 3.3. Neural results

As the selective attention task showed intervention-related performance differences, i.e., processing speed improvement on target trials, we analyzed EEG data for this specific task. Fig 4A shows ERSP time-frequency plots for visual electrodes (O1 and O2) time-locked to selective attention target stimuli. Peak target post-stimulus processing occurred in the 100−300ms time period across all sessions and subjects. Event-related synchronization (ERS) in the alpha frequency band (8−12 Hz), especially in the baseline pre-session that was reduced in the post-session (demarcated by red boxes in Fig 4A), dominated

**Table 4. Summary of cognitive outcomes obtained in both groups. Processing speed units are log(sec⁻¹) as the log inverse of the response time in seconds. Working memory (WM) span is out of 8 max item span and is normalized to 1. Length of the WM period is in seconds. Effect sizes of between group differences (Hedge's g) with lower and upper 95% confidence intervals and are also shown.**

| Outcomes | Mindfulness Intervention | | Control | | Intervention Effect Sizes |
|---|---|---|---|---|---|
| | Pre | Post | Pre | Post | |
| Selective Attention Processing Speed | 0.39±0.06 | 0.41±0.05 | 0.40±0.03 | 0.38±0.03 | 1.13 [0.01, 0.05] |
| Interference Processing Speed | 0.38±0.06 | 0.39±0.04 | 0.37±0.04 | 0.38±0.03 | 0.14 [−0.02, 0.03] |
| Working Memory Span | 0.47±0.36 | 0.63±0.32 | 0.51±0.31 | 0.55±0.30 | 0.35 [−0.09, 0.33] |
| Working Memory Period | 25.76±10.07 | 22.31±10.56 | 32.09±11.74 | 22.00±10.05 | 0.46 [−2.44, 15.72] |
| Emotion Bias Processing Speed | 0.35±0.05 | 0.36±0.04 | 0.35±0.03 | 0.35±0.03 | 0.00 [−0.02, 0.02] |

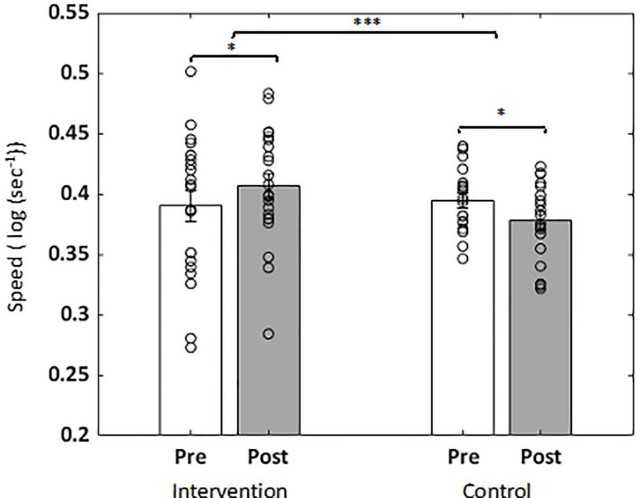

**Fig 3. Changes in Selective Attention Processing Speed by session and group.** Processing speed improved for the mindfulness intervention group and worsened for the control group. Individual data points are shown as scatterplot; bar length shows mean and error bars as standard error of mean *p<.05, ***p<.005.

peak post-stimulus neural activity. We source-localized this visual alpha activity and quantified the magnitude of source activity in visual ROIs (Fig 4B). Visual ROIs included left and right visual striate pericalcarine ROIs and extra-striate lingual, lateral occipital, cuneus and fusiform ROIs. The magnitude of visual alpha source activity was reduced at the post relative to pre-session in both groups such that the session x group rm-ANOVA showed a main effect of session ($F_{1,38}=6.93$, p=0.012) but no main effect of group (p>0.05) and no group x session interaction (p>0.05). However, the extent of post vs. pre visual alpha suppression was significantly correlated with the improvement in selective attention speed (partial Spearman correlation controlling for group, r=−0.32, p<0.05, Fig 4C). This correlation showed that individuals with greater increase in selective attention processing speed at the post- relative to pre-session showed greater visual source alpha suppression.

### 3.4. Experience feedback

These ratings were obtained at the end of the year by which time both groups had undergone the intervention. Average overall experience rating was very good to good (mean±standard deviation, 3.70±0.98) on a 5 point Likert scale with 5 being excellent. Table 5 shows the group specific ratings with no significant between-group differences. Additionally, 57% of all participants said "yes" in response to *As a result of this experience, do you expect your behavior to change in the future?*". We extracted themes

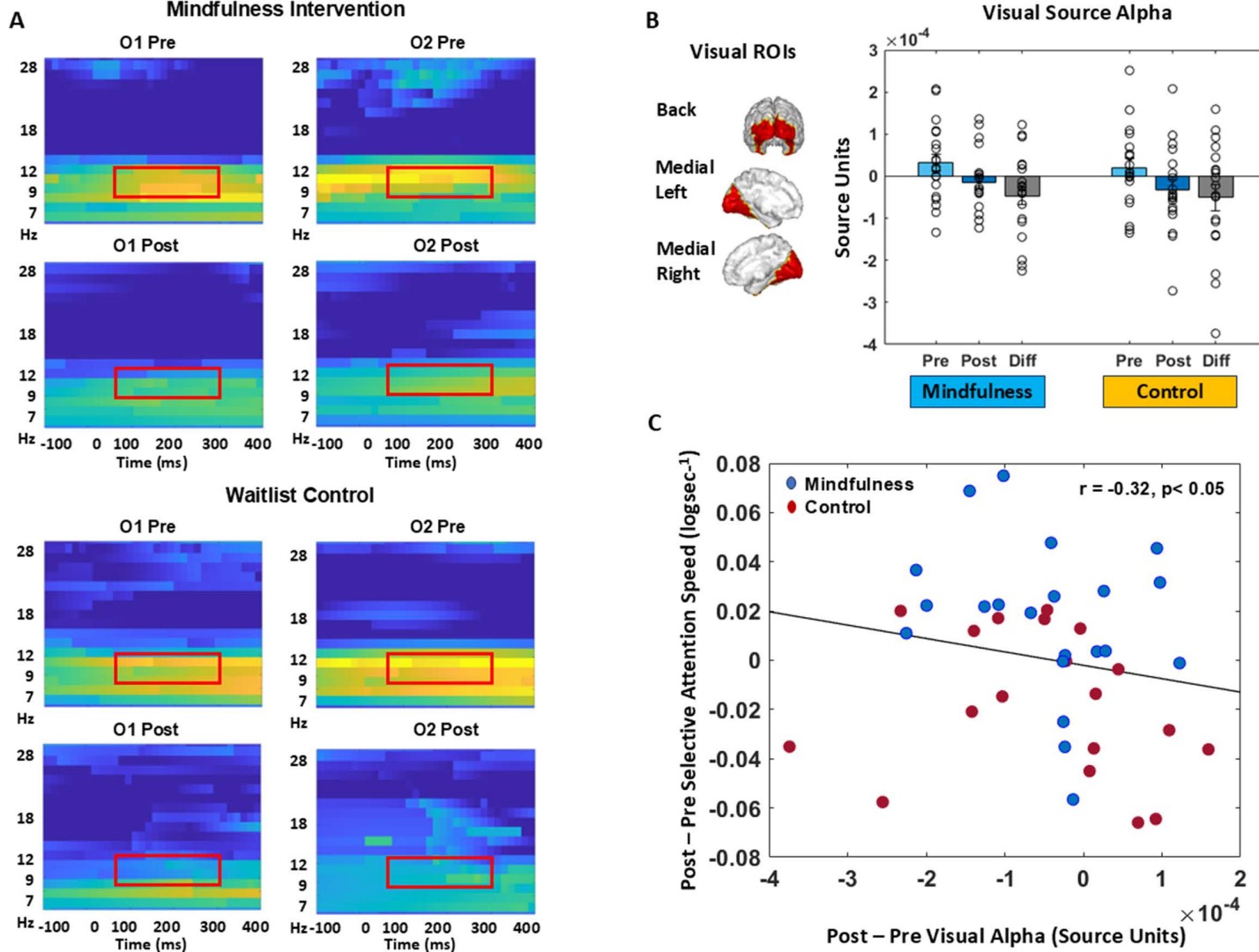

**Fig 4. Neural Activity on the Selective Attention Task by session and group. (A)** ERSP responses are shown in the mindfulness intervention group (top) and the control group (bottom) at occipital (O1:left and O2:right) electrodes at the pre- and post-sessions. Visual target event-related synchronization in the alpha band (8-12 Hz) is shown by red boxes. **(B)** Visual source ROIs are shown and the alpha activity in these ROIs is quantified at right. Individual data points are shown as scatterplot with bar length showing mean and error bars depicting standard error of mean. Both groups showed visual source alpha suppression at post- relative to pre-session. **(C)** Partial Spearman correlation controlling for group shows a significant inverse relationship between post minus pre change in selective attention processing speed vs. post minus pre change in visual alpha source activity.

for qualitative responses on what they particularly liked/learned from the program and/or what could be improved. These themes included attention to lifestyle factors (such as sleep and exercise as determinants of well-being), learning mindfulness and compassion, and that the study generated a sense of community. Some participants voiced a need for structural change to address the issues faced by medical scientists Proportions of these themes in each group are shown in Table 5; there were no group differences.

## 4. Discussion

This study investigated the outcomes of a brief quarter-long mindfulness intervention among M.D./Ph.D. trainees and graduates. Our goal was to enhance wellness in this cohort and address burnout. The program included three online

**Table 5. Summary of experience feedback outcomes obtained after both groups had undergone the intervention. Overall experience was rated on a 1-5 Likert Scale; expected behavior change as a result of the intervention had a yes/no response; the proportion of individuals providing a yes response are shown. The proportion of group members who brought up specific themes in the qualitative responses are also shown. There were no significant group differences in any feedback outcome variables.**

| Experience Feedback | Mindfulness Intervention | Control |
|---|---|---|
| Overall Experience (1:5 rating) | 3.76 ± 0.94 | 3.65 ± 1.01 |
| Expected Behavior Change | 0.47 | 0.65 |
| Attention to Lifestyle | 0.18 | 0.25 |
| Mindfulness | 0.29 | 0.4 |
| Compassion | 0.18 | 0.2 |
| Sense of Community | 0.29 | 0.1 |
| Need for Structural Change | 0.06 | 0.05 |

group sessions with psychological health experts who provided mindfulness skills and led discussions on how to manage work-life challenges. We evaluated both self-report and neuro-cognitive outcomes of the intervention in a randomized waitlist-controlled study design. Overall, the brief intervention was very feasible to deliver with all participants completing the intervention. Self-report outcomes of mindfulness, self-compassion, well-being and burnout showed no significant change, but neuro-cognitive assessments demonstrated that selective attention processing speed was improved. EEG-based neural results suggested that the improvement in selective attention speed may be related to on-task visual alpha suppression – a neural marker for suppression of distractions [120–124].

While the intervention had full participation and completion, reasons for not observing significant changes on self-report measures could be due to dosing/duration of the intervention or differences in style of the intervention compared to prior studies. For example, Krasner et al. (2009) implemented an 8-week intensive educational program (2.5 hours/week, 7-hour retreat) in mindfulness, communication, and self-awareness in primary care physicians. Pflugeisen et al., (2016) also implemented a mindfulness program in physicians over 8-weeks with weekly one-hour group teleconference coaching calls, weekly 5–7 minutes mindfulness training videos, and three 90-minutes in-person sessions. The intervention in our study had only three 90-minutes online group sessions over three months, and during the interim period participants were encouraged to continue their daily practice at home. Yet, notably, the feedback surveys from both groups that underwent the intervention suggested an overall positive experience with emergent themes of learning mindfulness, compassion, sense of community and attention to lifestyle factors. Of all the participants, 57% reported that they would engage in lifestyle behaviors that support well-being as a result of the intervention, which suggests long-term benefit that can be evaluated by future follow-up assessments.

We measured cognitive outcomes in the domains of selective attention, interference processing, working memory and emotion bias. These assessments were adaptive and with a threshold to maintain accuracy at ~80% so that ceiling effects would not occur with repeat administration. Cognitive outcomes showed a significant post- versus pre-intervention improvement in selective attention processing speed in the mindfulness intervention group, while the control group had a decrease in performance at their repeat session. Such intervention-driven modulation of visual attention is consistent with prior studies of meditation-based psychosocial intervention [125–127]. Digital meditation practice in young adults and children over a 6–8 week period have also shown improvement in selective attention skills [72,128]; in those studies, worse performance with repeat test administration without intervention could be associated with boredom or because the rejuvenating meditative intervention was not offered to the control group. The pattern of current findings fits with the energetic-resource model, which proposes that the state of resource depletion is reflected in the gradual decline of task performance speed over time [11]. Hence, we interpret the observed improvement in processing speed in the intervention

group might be due the replenishment of attentional resources by the practice of mindfulness. We also note that although we evaluated a battery of cognitive measures, selective attention results survived multiple comparisons correction across all tests and had large bias corrected between group effect size (Hedges' g = 1.13). Nonetheless, our small study sample urges the need for follow-up studies with larger sample sizes.

Our prior digital meditation studies also found improvements in interference processing and interoceptive attention, which were potentially not observed here due to differences in intervention dose and style [54,72]. Indeed, a review of cognitive effects of mindfulness trainings by Chiesa et al [62] concluded that cognitive functions show differential sensitivity to mindfulness-based interventions governed by varying factors in empirical studies, such as intervention study design, duration and participating populations. It should also be noted that the intervention in the current study incorporated a broader spectrum of tools beyond mindfulness, such as compassion, inspiration, strength reflection, goal-setting and fear management, among others.

At the neural level, visual alpha oscillations were suppressed at the post- relative to pre-session that maybe linked with changes in selective attention speed as shown by the negative correlation between these measures. Alpha band oscillations are considered a sensory suppression mechanism in visual selective attention [120–124]. These studies have shown that sensory visual alpha suppression is controlled by top–down signals from frontoparietal attention networks. Thus, suppression of visual alpha band power following a stimulus is thought to measure attentional selection. Selective attention requires one to simultaneously attend to goal-relevant target stimuli while ignoring goal-irrelevant non-target stimuli. Alpha suppression for target stimuli (but not for non-targets) can facilitate selection. Post-stimulus visual processing usually also shows theta/alpha band ERS related to underlying spiking of visual sensory neurons [129], which is distinct from alpha suppression driven by top-down modulation. Here, we found that target stimuli had greater visual alpha suppression at the post-intervention session, suggesting improved attention selection. However, both groups at the group-average level showed this effect, which might be attributed to practice effects with repeat exposure in the control group. To investigate individual level effects, i.e., whether this plasticity in visual alpha may be relevant to change in selective attention performance speed, we performed a partial correlation controlling for group as a factor. We found a significant inverse correlation between improvement in selective attention processing speed and negative change in visual alpha. This result suggests that visual alpha suppression may underlie the individual-level improvement in cognitive performance. The relatively small effect size of the neural modulation suggests future studies need to replicate this outcome with larger sample sizes and using stronger intervention and control groups.

To the best of our knowledge, this is the first randomized, controlled study of a mindfulness-integrated intervention conducted with medical scientists and that assessed self-report measures alongside neuro-cognitive outcomes of selective attention, working memory, interference processing and emotional bias processing. However, this study has several limitations. Firstly, the study uses a passive (waitlist) control condition, which limited our ability to disentangle the treatment effects from non-specific effects, including demand characteristics, social desirability, placebo, expectation, motivation, and contact with the group members and leaders. This limitation is quite common in mindfulness studies and has been pointed out in several review papers [127,130–133]. For example, Davidson and Kaszniak (2015) noted that unlike pharmacological interventions, implementing a double-blind placebo-controlled design in meditation-based interventions is nearly impossible because participants can easily discern if they are assigned to a meditation condition, making it quite challenging to keep them unaware of the nature of the intervention. Additionally, Tang et al (2015) pointed out that participants often join the intervention with expectation and intention that may also confound the effect of the actual program. Given tight resource constraints, we were unable to design and implement an active control group in the current study. The second major limitation of the study was the small sample size of 43 participants, especially when we evaluated several self-report and cognitive measures albeit with statistical corrections applied for multiple comparisons. The third limitation of the study was potentially having too small an intervention dose, although feasible within the busy schedules of medical scientists, to show any detectable improvements particularly apparent in the null self-report outcomes. We note

that prior interventions conducted with physician cohorts have used larger intervention doses such as 8-week intensive education programs [32,34,36–38]. Thus, overall, the present study can be viewed as a pilot for a future study with larger sample size, an active control group, and potentially greater dose, i.e., frequency and duration of mindfulness coaching sessions to produce robust outcomes.

In addition to the low sample size of the pilot study, we further acknowledge that our study participants had low burnout on average, which may explain some of the null findings especially for the self-report measures. It is also possible that heterogeneity in our cohort of M.D./Ph.D. trainees and graduates may have led to variability in outcomes and contributed to null results. Hence, future work may consider focusing on a high burnout cohort that is also uniform with regard to the training stage of study participants. Notably, prior work has also suggested that organization-directed workplace interventions could be effective for addressing burnout [134]. The need for structural change was noted in the feedback survey by ~5% of our participants. Hence, future studies may explore how mindfulness-integrated interventions can be deployed at a larger scale within an organizational framework that seeks to enhance physician well-being.

In conclusion, this randomized, controlled study sought to examine the effects of a mindfulness-integrated psychological intervention on improvement of subjective well-being related behaviors, cognitive functioning and associated electrophysiological dynamics. The inclusion of EEG-based cognitive assessments is a strength of this study. Despite the low intensity training, we found significant improvements in selective attention in the intervention group; this cognitive plasticity was correlated with neural alpha suppression in visual cortex. Participants rated their intervention experience as very good, and a majority of participants planned to change their well-being related behaviors as a result of the intervention. Participants liked that the intervention focused on lifestyle factors, mindfulness and compassion, and that the study offered a sense of community. Based on these findings, we believe that a larger scale implementation of such an intervention applied with greater intensity and integrated within the education curriculum may be useful for increasing well-being among physician scientist trainees and graduates. Overall, we posit that the well-being of our physicians and physician scientist workforce should be a pro-active and vital component of training to reduce burnout in this community and mitigate its deleterious consequences for the healthcare system.

## Supporting information

**S1 File. CONSORT 2010 checklist.**
(DOC)

**S2 File. Study protocol updated.**
(DOC)

## Author contributions

**Conceptualization:** Suzanna R. Purpura, James K. Manchanda, Iris Garcia-pak, Dhakshin S. Ramanathan, Dawna Chuss, Deborah T. Rana, Ellen Beck, Paul A. Insel, Jyoti Mishra.

**Data curation:** Satish Jaiswal, Jason Nan, Suzanna R. Purpura, James K. Manchanda, Dhakshin S. Ramanathan, Deborah T. Rana, Ellen Beck, Paul A. Insel, Neil C. Chi, David M. Roth, Hemal H. Patel, Jyoti Mishra.

**Formal analysis:** Satish Jaiswal, Jason Nan, Jyoti Mishra.

**Funding acquisition:** David M. Roth, Hemal H. Patel, Jyoti Mishra.

**Investigation:** Dhakshin S. Ramanathan, Dawna Chuss, Deborah T. Rana, Ellen Beck, Paul A. Insel, Neil C. Chi, David M. Roth, Jyoti Mishra.

**Methodology:** Satish Jaiswal, Jyoti Mishra.

**Project administration:** Satish Jaiswal, Dhakshin S. Ramanathan, David M. Roth, Hemal H. Patel, Jyoti Mishra.

**Resources:** Satish Jaiswal, Jason Nan, Dhakshin S. Ramanathan, Deborah T. Rana, David M. Roth, Hemal H. Patel, Jyoti Mishra.

**Software:** Satish Jaiswal, Jason Nan, Dhakshin S. Ramanathan, Jyoti Mishra.

**Supervision:** Suzanna R. Purpura, James K. Manchanda, Dhakshin S. Ramanathan, David M. Roth, Jyoti Mishra.

**Validation:** Satish Jaiswal, Dhakshin S. Ramanathan, Jyoti Mishra.

**Visualization:** Jyoti Mishra.

**Writing – original draft:** Satish Jaiswal, Iris Garcia-pak, Dhakshin S. Ramanathan, Jyoti Mishra.

**Writing – review & editing:** Satish Jaiswal, Jyoti Mishra.

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
