## [Decision Letter · Decision Letter 0]

29 May 2025

PONE-D-25-15885Brief Mindfulness Coaching Enhances Selective Attention in Medical Scientists: A Pilot StudyPLOS ONE

Dear Dr. Jaiswal,

Thank you for submitting your manuscript to PLOS ONE. After careful consideration, we feel that it has merit but does not fully meet PLOS ONE’s publication criteria as it currently stands. Therefore, we invite you to submit a revised version of the manuscript that addresses the points raised during the review process. I was able to secure two expert reviews, both of which are now available for your consideration. As you will see, both reviewers raise several substantive critiques. However, their evaluations are constructive and include concrete suggestions for improvement. I encourage you to review these comments carefully and to address the raised issues thoroughly in your revision, particularly with respect to how the purpose and implications of your study are framed, especially in its role as a pilot study. I would like to invite you to submit a revised version of your manuscript. Please include a detailed response letter that addresses each reviewer comment in a point-by-point format, indicating clearly how the manuscript has been revised in response. I look forward to receiving your revised submission.Please submit your revised manuscript by Jul 12 2025 11:59PM. If you will need more time than this to complete your revisions, please reply to this message or contact the journal office at plosone@plos.org . Please include the following items when submitting your revised manuscript:

We look forward to receiving your revised manuscript.

Kind regards,

Michael B. Steinborn, PhD

Section Editor

PLOS ONE

 [This work was supported by a grant from the National Institutes of Health, NIH/NIGMS 1T32GM121318-01 (DMR, HHP), involved UCSD-MSTP trainees supported by NIH/NIGMS 5T32GM007198, and a Research Career Scientist Award from the Veterans Administration (BX005229 to HHP).]. 

[This work was supported by a grant from the National Institutes of Health, NIH/NIGMS 1T32GM121318-01 (DMR, HHP), involved UCSD-MSTP trainees supported by NIH/NIGMS 5T32GM007198, and a Research Career Scientist Award from the Veterans Administration (BX005229 to HHP). We thank Alankar Misra for software development of the BrainE platform and UCSD students who assisted with data collection. The BrainE software is copyrighted for commercial use (Regents of the University of California Copyright #SD2018-816) and free for research and educational purposes.]

 [This work was supported by a grant from the National Institutes of Health, NIH/NIGMS 1T32GM121318-01 (DMR, HHP), involved UCSD-MSTP trainees supported by NIH/NIGMS 5T32GM007198, and a Research Career Scientist Award from the Veterans Administration (BX005229 to HHP).]

6. In the online submission form, you indicated that your data will be submitted to a repository upon acceptance.  We strongly recommend all authors deposit their data before acceptance, as the process can be lengthy and hold up publication timelines. Please note that, though access restrictions are acceptable now, your entire minimal  dataset will need to be made freely accessible if your manuscript is accepted for publication. This policy applies to all data except where public deposition would breach compliance with the protocol approved by your research ethics board. If you are unable to adhere to our open data policy, please kindly revise your statement to explain your reasoning and we will seek the editor's input on an exemption.

Reviewers' comments:

Reviewer's Responses to Questions

**Comments to the Author**

1. Is the manuscript technically sound, and do the data support the conclusions?

Reviewer #1: Yes

Reviewer #2: No

2. Has the statistical analysis been performed appropriately and rigorously? 

Reviewer #1: Yes

Reviewer #2: Yes

3. Have the authors made all data underlying the findings in their manuscript fully available?

Reviewer #1: Yes

Reviewer #2: Yes

4. Is the manuscript presented in an intelligible fashion and written in standard English?

Reviewer #1: Yes

Reviewer #2: Yes

5. Review Comments to the Author

Reviewer #1: Dear Editor,

Thanks for the opportunity to submit my review report titled “Brief Mindfulness Coaching Enhances Selective Attention in Medical Scientists: A Pilot Study “the hypothesis that brief mindfulness coaching (three 1.5-hour online group sessions over 12 weeks) can benefit medical scientists”

Here are my comments: The authors have included a section that captures detailed power analysis procedures conducted to arrive at a sample size of 43. Therefore, the study is powered to test the proposed hypothesis.

The intervention and control groups have been clearly defined, and detailed highlights of how the intervention group benefitted from the Brief Mindfulness Coaching have been provided.

Statistical Analysis: The use of the Chi-square test of independence for assessing associations between two categorical variables (gender and ethnicity), especially when most of the expected frequencies were less than 5 from Table 1, was not appropriate. Ideally, Fisher's Exact test would have been better. Using a non-parametric Wilcoxon rank sum test for age comparisons between intervention and control groups was appropriate, but the authors should include why the non-parametric approach was adopted. Was it because there was a violation of normality and equal variance assumption required for a parametric two-sample t-test for age? How they tested for normality and equal variance is also missing under the Demographic characteristics section. They can equally provide a summary of how key statistical assumptions were tested to cater for all variables.

Please add a section called outcome measures (primary and secondary outcome measures) to provide proper context for the choice of statistical methods. You lumped all of them under data analysis. Kindly titled them outcome measures

The use of repeated measure ANOVA and Kruskal Wallis tests were appropriate.

My major concern. This is an intervention study, but the results presented in Tables 2 and 3 do not show the intervention's true effect. Authors only presented pre-post for intervention and control groups, but the true effect size, which is a difference in difference, was not presented, making it difficult to determine the true effect of the intervention. That is the change in the intervention group minus the change in the control, including the corresponding confidence interval, would represent the true effect size, which is the difference in differences estimate.

Reviewer #2: Thank you for the opportunity to review this manuscript. The focus on supporting medical scientists, who are often under significant cognitive and emotional demands, through an abbreviated mindfulness intervention is both timely and important. The integration of EEG-based cognitive assessments alongside self-reported outcomes adds scientific rigor to the work, particularly in evaluating selective attention changes.

That said, several aspects of the manuscript would benefit from further clarification and refinement to enhance its clarity, coherence, and impact. The rationale for the intervention design—including selection and ordering of mindfulness practices—could be made more explicit, and the use of extensive terms and quotations from external sources could be streamlined to improve readability. There is real potential here to contribute meaningfully to the literature on mindfulness interventions in high-stress training environments. With improved structural clarity and a more grounded theoretical justification for intervention components, this manuscript could be a valuable contribution to the field.

I appreciate the authors’ thoughtful engagement with this important topic and look forward to future developments in this line of work. Please see the attached comments.

6. PLOS authors have the option to publish the peer review history of their article (what does this mean? ). If published, this will include your full peer review and any attached files.

**Do you want your identity to be public for this peer review?** For information about this choice, including consent withdrawal, please see our Privacy Policy .

Reviewer #1: No

Reviewer #2: No

---

## [Author Response · Author response to Decision Letter 1]

9 Jul 2025

Editorial Comments

Response. We have formatted the manuscript per PLOS ONE’s style requirements.

Response. We have now made these sections consistent regarding funding.

[This work was supported by a grant from the National Institutes of Health, NIH/NIGMS 1T32GM121318-01 (DMR, HHP), involved UCSD-MSTP trainees supported by NIH/NIGMS 5T32GM007198, and a Research Career Scientist Award from the Veterans Administration (BX005229 to HHP).].

Response. We have now stated that “The funders had no role in study design, data collection and analysis, decision to publish, or preparation of the manuscript.”

[This work was supported by a grant from the National Institutes of Health, NIH/NIGMS 1T32GM121318-01 (DMR, HHP), involved UCSD-MSTP trainees supported by NIH/NIGMS 5T32GM007198, and a Research Career Scientist Award from the Veterans Administration (BX005229 to HHP). We thank Alankar Misra for software development of the BrainE platform and UCSD students who assisted with data collection. The BrainE software is copyrighted for commercial use (Regents of the University of California Copyright #SD2018-816) and free for research and educational purposes.]

[This work was supported by a grant from the National Institutes of Health, NIH/NIGMS 1T32GM121318-01 (DMR, HHP), involved UCSD-MSTP trainees supported by NIH/NIGMS 5T32GM007198, and a Research Career Scientist Award from the Veterans Administration (BX005229 to HHP).]

Response. We have removed funding related text from the manuscript. The funding statement is correct.

Response. Thank you for pointing this out. We have now ensured that the Ethics statement only appears in the Methods > Participants section.

6. In the online submission form, you indicated that your data will be submitted to a repository upon acceptance. We strongly recommend all authors deposit their data before acceptance, as the process can be lengthy and hold up publication timelines. Please note that, though access restrictions are acceptable now, your entire minimal dataset will need to be made freely accessible if your manuscript is accepted for publication. This policy applies to all data except where public deposition would breach compliance with the protocol approved by your research ethics board. If you are unable to adhere to our open data policy, please kindly revise your statement to explain your reasoning and we will seek the editor's input on an exemption.

Response. The dataset generated and analyzed in the current study is available from the datadryad.org repository: https://datadryad.org/dataset/doi:10.5061/dryad.zpc866tfb

Please note that this link will be active once the study is published.

Review Comments to the Author

Reviewer #1: Dear Editor,

Thanks for the opportunity to submit my review report titled “Brief Mindfulness Coaching Enhances Selective Attention in Medical Scientists: A Pilot Study “the hypothesis that brief mindfulness coaching (three 1.5-hour online group sessions over 12 weeks) can benefit medical scientists”

Here are my comments:

1. The authors have included a section that captures detailed power analysis procedures conducted to arrive at a sample size of 43. Therefore, the study is powered to test the proposed hypothesis.

2. The intervention and control groups have been clearly defined, and detailed highlights of how the intervention group benefitted from the Brief Mindfulness Coaching have been provided.

Statistical Analysis: The use of the Chi-square test of independence for assessing associations between two categorical variables (gender and ethnicity), especially when most of the expected frequencies were less than 5 from Table 1, was not appropriate.

Ideally, Fisher's Exact test would have been better. Using a non-parametric Wilcoxon rank sum test for age comparisons between intervention and control groups was appropriate, but the authors should include why the non-parametric approach was adopted. Was it because there was a violation of normality and equal variance assumption required for a parametric two-sample t-test for age? How they tested for normality and equal variance is also missing under the Demographic characteristics section. They can equally provide a summary of how key statistical assumptions were tested to cater for all variables.

Please add a section called outcome measures (primary and secondary outcome measures) to provide proper context for the choice of statistical methods. You lumped all of them under data analysis. Kindly titled them outcome measures. The use of repeated measure ANOVA and Kruskal Wallis tests were appropriate.

Response. We thank the Reviewer for their thoughtful comments regarding the statistical approach in our study. Following the Reviewer’s suggestions we have updated the ‘Data Analysis’ section as ‘Outcome Measures and Data Analysis’. We have also allocated separate subsections for each outcome measures (Demographics, Self-report, Cognitive Assessment and Neural Assessment).

To address the Reviewer’s concerns regarding the demographic characteristics we would like to clarify that our study did not intend to find any association between categorical variables such as gender and ethnicity. Our decision to conduct Chi-square testing was simply to ensure that the two groups did not differ in either their distribution of gender or ethnicity. Since age was a non-categorical variable, we used the Wilcoxon rank sum test to compare age between groups.

We have further addressed the Reviewer’s concern by adding the following statements under the heading ‘Outcome Measures and Data Analysis’:

All the study measures were tested for normality using the Levene’s test. The normally distributed data were evaluated with parametric repeated measure analysis of variance (rm-ANOVA) for group x session interactions, while non-normal data were evaluated using the non-parametric equivalent of ANOVA, i.e., Kruskal-Wallis (KW) test (McKight & Najab, 2010). For any significant ANOVA results, post-hoc analyses were conducted within each group using either paired t-tests for normal data or the non-parametric equivalent Wilcoxon signed-rank test for non-normal data, and between-group effects in the ANOVA were further interrogated using either the two-sample t-test for normal data or otherwise the non-parametric Wilcoxon rank sum test was performed.

My major concern. This is an intervention study, but the results presented in Tables 2 and 3 do not show the intervention's true effect. Authors only presented pre-post for intervention and control groups, but the true effect size, which is a difference in difference, was not presented, making it difficult to determine the true effect of the intervention. That is the change in the intervention group minus the change in the control, including the corresponding confidence interval, would represent the true effect size, which is the difference in differences estimate.

Response. We thank the Reviewer for raising this concern. Following the Reviewer’s suggestion we have revised Tables 2 and 3 to show the confidence intervals and effect sizes based on the change in the intervention group minus the change in the control group.

Reviewer #2:

Thank you for the opportunity to review this manuscript. The focus on supporting medical scientists, who are often under significant cognitive and emotional demands, through an abbreviated mindfulness intervention is both timely and important. The integration of EEG-based cognitive assessments alongside self-reported outcomes adds scientific rigor to the work, particularly in evaluating selective attention changes.

That said, several aspects of the manuscript would benefit from further clarification and refinement to enhance its clarity, coherence, and impact. The rationale for the intervention design—including selection and ordering of mindfulness practices—could be made more explicit, and the use of extensive terms and quotations from external sources could be streamlined to improve readability. There is real potential here to contribute meaningfully to the literature on mindfulness interventions in high-stress training environments. With improved structural clarity and a more grounded theoretical justification for intervention components, this manuscript could be a valuable contribution to the field.

Response. We have now edited the manuscript for greater clarity, coherence and impact. The rationale for the intervention is now provided in the first paragraph of the intervention, and selection and ordering of practices has been explicitly stated. Extensive terms and quotes have also been streamlined.

I appreciate the authors’ thoughtful engagement with this important topic and look forward to future developments in this line of work. Please see the attached comments.

Page 5–Line 8: What is BrainE? Please define this term clearly upon first use. If it refers to a proprietary tool or software, consider presenting it as: “we have validated the reliability of BrainE, a [brief description, e.g., cognitive assessment] platform,” rather than “the reliability of this BrainE platform.”

Response. We thank the Reviewer for their comment. In this revision we have described BrainE with greater clarity as below:

“here we implemented a scalable platform, the Brain Engagement (BrainE©) tool, developed for free academic use by the Neural Engineering and Translation Labs (NEATLabs), for measuring core aspects of cognition. The BrainE© tool delivers standard cognitive assessments synchronized with neural recordings using electroencephalography (EEG) (Balasubramani et al., 2021; Misra et al., 2018). We have validated the reliability of BrainE© and shown its utility in measuring cognitive changes across the lifespan (Balasubramani, Diaz-Delgado, et al., 2022; Balasubramani, Walke, et al., 2022; G. Grennan et al., 2021, 2022; Mo et al., 2023; Ramanathan et al., 2024). We have also demonstrated its relevance in predicting mental health, well-being and burnout (G. K. Grennan et al., 2023; Kato et al., 2022; Nan et al., 2022, 2024; Shah et al., 2021).

3. 5–34: Using EEG measures is a large part of this study; could the authors clarify whether this constitutes a truly novel approach specifically in the context of interventions targeting health professionals? Even if it’s not the first study to use EEG measures in this capacity, the inclusion of imaging techniques is great.

Response. We thank the Reviewer for this comment, and have modified the text for greater clarity as below:

“EEG measures can provide insights into neural plasticity induced by the intervention (Anguera et al., 2013; Mishra et al., 2014, 2015, 2020, 2021), which has not been applied in the context of interventions for the healthcare community apart from a couple of recent studies (Jaiswal et al., 2024, White et al., 2024).”

4. 5–42: The acronym “MSTP” is defined only at the end; it would improve clarity to define it upon first use.

Response. We thank the Reviewer for this suggestion. The MSTP acronym has now been defined upon its first use as “participants were recruited from the University of California San Diego Medical Scientist Training Program (UCSD-MSTP)”.

4. 7–Figure 1:

- What were the inclusion criteria for participants?

Response. We have stated the inclusion criteria in the revised manuscript, “Inclusion criteria were current enrollment in the UCSD MSTP program or graduates of MD-PhD program.”

- Consider whether this figure is the most effective use of space. Much of the information is redundant with the adjacent text or could be simplified. For example, listing the allocation group along with the number who did not receive the intervention may be unnecessary when the box is already labeled “Allocation to Waitlist Control Group (n = X).” It may be more useful to detail participant characteristics within each group (e.g., pre-clinical MSTP).

Response. We totally agree with the Reviewer’s comment. However, it was a mandatory requirement requested by the Journal’s editorial team to provide a Consort Flowchart as Figure 1.

- There appears to be a discrepancy: the figure indicates that 0 participants were excluded from analysis, but the text on line 6–23 states that some data were missing and excluded. Please clarify this inconsistency.

Response. After the enrollment stage, no participants were excluded from allocation and analysis stages. For transparency, we have explicitly detailed how many participants had their data missing on what measures in the manuscript text. To avoid confusion, we have deleted the following sentence, “Missing data were excluded from analyses.” The final number of participants in both groups represents correct number of participants whose data were analyzed.

Were there any subgroup differences? (e.g., was one effect stronger for junior faculty vs. graduate students)?

Response. No there were no subgroup differences.

That said, several aspects of the manuscript would benefit from further clarification and r

---

## [Decision Letter · Decision Letter 1]

30 Jul 2025

Brief Mindfulness Coaching Enhances Selective Attention in Medical Scientists: A Pilot Study

PONE-D-25-15885R1

Dear Dr. Jaiswal,

We’re pleased to inform you that your manuscript has been judged scientifically suitable for publication and will be formally accepted for publication once it meets all outstanding technical requirements.

Editor comment: The two reviewers have now evaluated your revised manuscript. As you can see from their comments, both consider the paper ready for publication. They confirm that all previously raised issues have been addressed satisfactorily. R2 suggests a few minor corrections. I believe these can be implemented during the final preparation phase and do not require a further round of review. I am therefore pleased to accept your manuscript for publication. Congratulations on this outcome. 

Kind regards,

Michael B. Steinborn, PhD

Section Editor

PLOS ONE

Additional Editor Comments (optional):

Reviewers' comments:

Reviewer's Responses to Questions

**Comments to the Author**

1. If the authors have adequately addressed your comments raised in a previous round of review and you feel that this manuscript is now acceptable for publication, you may indicate that here to bypass the “Comments to the Author” section, enter your conflict of interest statement in the “Confidential to Editor” section, and submit your "Accept" recommendation.

Reviewer #1: All comments have been addressed

Reviewer #2: (No Response)

2. Is the manuscript technically sound, and do the data support the conclusions?

Reviewer #1: Yes

Reviewer #2: Yes

3. Has the statistical analysis been performed appropriately and rigorously? 

Reviewer #1: Yes

Reviewer #2: Yes

4. Have the authors made all data underlying the findings in their manuscript fully available?

Reviewer #1: Yes

Reviewer #2: Yes

5. Is the manuscript presented in an intelligible fashion and written in standard English?

Reviewer #1: Yes

Reviewer #2: Yes

6. Review Comments to the Author

Reviewer #1: All my previous comments have been duly addressed by the authors. They also provided further clarification where necessary.

Reviewer #2: Thank you for your thoughtful and comprehensive revisions in response to my initial feedback. I appreciate the care taken to clarify the intervention rationale, streamline terminology, and better define methodological components such as the use of EEG and participant grouping. The manuscript is now much clearer and more cohesive.

I have just a few remaining points for clarification or consideration before finalizing:

1. Table Mapping Practices to Outcomes

Thank you for elaborating on the rationale for not including a table linking mindfulness practices to targeted outcomes. I understand your concern regarding oversimplification, and the revised text provides helpful context. That said, even a brief summary figure or schematic—indicating general outcome domains (e.g., attention, emotion regulation)—might support reader understanding while preserving nuance.

2. Novelty of EEG Use

The inclusion of EEG-based cognitive assessments is a strength of this study. To further underscore the contribution, consider framing the novelty of this application more explicitly.

7. PLOS authors have the option to publish the peer review history of their article (what does this mean? ). If published, this will include your full peer review and any attached files.

**Do you want your identity to be public for this peer review?** For information about this choice, including consent withdrawal, please see our Privacy Policy .

Reviewer #1: No

Reviewer #2: No

---

## [Editor Report · Acceptance letter]

PONE-D-25-15885R1

PLOS ONE

Dear Dr. Jaiswal,

I'm pleased to inform you that your manuscript has been deemed suitable for publication in PLOS ONE. Congratulations! Your manuscript is now being handed over to our production team.

Kind regards,

on behalf of

Dr. Michael B. Steinborn

Section Editor

PLOS ONE